# Antecedents and Outcomes of Work Engagement among Psychiatric Nurses in Japan

**DOI:** 10.3390/healthcare11030295

**Published:** 2023-01-18

**Authors:** Yuichi Kato, Rie Chiba, Akihito Shimazu, Yuta Hayashi, Takuya Sakamoto

**Affiliations:** 1Hyogo Prefectural Hyogo Mental Health Center, Kobe 651-1242, Japan; 2Department of Nursing, Graduate School of Health Sciences, Kobe University, Kobe 654-0142, Japan; 3Faculty of Policy Management, Keio University, Kanagawa 252-0882, Japan; 4Department of Nursing, Osaka Psychiatric Medical Center, Osaka 541-8567, Japan

**Keywords:** job crafting, nursing practice environment, psychiatric nurses, strength-oriented care attitudes, work engagement

## Abstract

While previous studies have examined antecedents and outcomes of work engagement among general nurses, studies among psychiatric nurses remain limited. This study aimed to explore the antecedents (i.e., job crafting and nursing practice environment) and outcomes (i.e., strength-oriented care attitudes, mental health, and turnover intention) of work engagement among psychiatric nurses in Japan. This cross-sectional study included 309 nurses from three psychiatric hospitals in Japan (valid response rate: 60.4%). Data collection using the self-administered questionnaire took place from July to August 2021. We performed Structural Equation Modeling to examine the directional relationships among variables. Job crafting (β = 0.57, *p* < 0.01) and nursing practice environment (β = 0.23, *p* = 0.01) exhibited positive effects on work engagement. Work engagement had positive effects on strength-oriented care attitudes (β = 0.15, *p* = 0.04) and mental health (β = 0.37, *p* < 0.01) as well as negative effects on intention to resign from their profession as a nurse (β = −0.17, *p* = 0.01). Job crafting and a healthier nursing practice environment could help enhance work engagement. Higher work engagement could contribute to improving strength-oriented care attitudes, mental health, and intention to resign from their profession as a nurse.

## 1. Introduction

The increased prevalence of mental illness has resulted in the growing demand for high-quality mental health care for patients of all ages [1]. Psychiatric nurses comprise the largest workforce among mental health professionals and provide close care to patients and their family members. Their job performance is directly related to quality of care in mental health services. However, they can often experience higher levels of emotional exhaustion and lower levels of personal accomplishment due to work-related stressors, such as physical, psychological, or verbal violence from patients, difficulties in nurse–patient relationships, and excessive workload [2,3,4]. These stressors experienced by psychiatric nurses could deteriorate job performance as well as mental health and turnover intention, leading to poor quality of care [5,6].

Work engagement, defined as “a positive, fulfilling, and work-related state of mind that is composed of (a) vigor, (b) dedication, and (c) absorption [7]”, has received attention in the fields of nursing [8]. Vigor means high levels of energy and mental resilience during work. Dedication refers to a sense of significance, enthusiasm, and pride. Absorption means complete concentration in one’s work [7]. In general, engaged employees have high levels of energy and better mental health [9]. While some empirical studies in general nurses showed higher work engagement could lead to high-quality care for patients [10,11,12,13,14], studies including psychiatric nurses remain limited [15,16]. Compared to general nurses, psychiatric nurses can often experience stressful situations more frequently, such as violence from patients, stress related to patient–nurse interpersonal relationships, and encountering suicidal attempts and self-harm [3]. According to past studies, such job characteristics may result in some differences in antecedents and outcomes of work engagement between psychiatric nurses and general nurses [4,5]. Further investigation is needed to explore the related factors of work engagement in psychiatric nurses.

Mental health care in recent years places importance on supporting the utilization and development of mentally ill patients’ preferences, abilities, hopes, and experiences, as well as the available resources and opportunities in their environments, which are considered “strengths” [17]. The strength-oriented care leads to many benefits for patients with mental illness, such as reduced length of hospitalization and improved adherence to medication as well as improved quality of life and life satisfaction [18,19]. However, no research examined the relationship between work engagement and strength-oriented care to date. Exploring this association among psychiatric nurses may provide great insights into the strategies for enhancing strength-oriented care attitudes, leading to high-quality mental health care.

Job crafting, defined as “the cognitive and physical change that individuals make in the task or relational boundaries of their work [20],” is proactive behaviors and cognitions that enhance their own resources. It consists of three components: changing the cognitive task boundaries (cognitive crafting), changing the job’s boundaries (task crafting), and changing the relational boundaries (relational crafting) [20]. These three components are theoretically responsible for helping employees have a positive attitude toward their work, enhancing work engagement [21,22]. However, no previous studies have empirically clarified the relationship between job crafting and work engagement among psychiatric nurses.

Nursing practice environment is defined as “the organizational characteristics of work settings that promote or inhibit nursing practice [23].” It is composed of (a) nurse participation in hospital affairs, (b) supervisors’ leadership and supports, (c) staffing and resource adequacy, (d) collegial nurse–physician relationship, and (e) nursing foundations for quality of care [23,24]. A supportive work environment for nursing practice could contribute to fostering nurses’ willingness to dedicate more effort into their work, leading to higher levels of work engagement through enhancing extrinsic motivation [15]. However, few studies have examined the relationship between nursing practice environment and work engagement among psychiatric nurses.

Past studies investigating outcomes of work engagement have reported that highly engaged nurses could tend to have better mental health [25] and lower turnover intention [11,26]. However, these studies did not include psychiatric nurses. It remains unclear whether work engagement could contribute to the improvement of these work-related outcomes among psychiatric nurses.

### 1.1. Aim

This study aimed to examine the antecedents and outcomes of work engagement among psychiatric nurses in Japan.

### 1.2. Conceptual Framework and Hypotheses in This Study

The Job Demands–Resources model (JD-R model) has routinely been adopted as a theoretical framework to delineate the antecedents and outcomes of work engagement [21]. Previous studies have also suggested that the JD-R model can be used to comprehensively understand work engagement and related factors among nurses [27,28]. According to this model, job resources (e.g., nursing practice environment) positively stimulate work engagement and improve work-related outcomes (e.g., job performance, mental health, and turnover intention) through a motivational process [21]. Job resources mean “physical, psychosocial, or organizational resources in the workplace for achieving personal work goals, reducing job stress, or facilitating personal growth and development [21]”. In addition, job crafting theoretically acts to improve job resources and then enhances work engagement in this model [21].

In this study, job crafting and nursing practice environment were considered as antecedents of work engagement. Strength-oriented care attitudes, mental health, and turnover intention (intention to leave current workplace and intention to resign from their profession as a nurse) were considered as outcomes of work engagement. Thus, we set the following hypotheses (Figure 1a–d), based on the JD-R model.

#### 1.2.1. Antecedents of Work Engagement

**Hypothesis 1.1.** 
*Job crafting has a direct and positive effect on work engagement.*


**Hypothesis 1.2.** 
*Job crafting has an indirect and positive effect on work engagement mediated by job resource (i.e., nursing practice environment).*


**Hypothesis 2.** 
*Nursing practice environment has a direct and positive effect on work engagement.*


#### 1.2.2. Outcomes of Work Engagement

**Hypothesis 3.1.** 
*Work engagement has a direct and positive effect on strength-oriented care attitudes (Figure 1a).*


**Hypothesis 3.2.** 
*Work engagement has a direct and positive effect on mental health (Figure 1b).*


**Hypothesis 3.3.** 
*Work engagement has a direct and negative association with intention to leave current workplace (Figure 1c).*


**Hypothesis 3.4.** 
*Work engagement has a direct and negative association with intention to resign from their profession as a nurse (Figure 1d).*


#### 1.2.3. Mediating Effects of Work Engagement

In the JD-R model [21], work engagement theoretically plays a role in mediating the relationships between job resources and work-related outcomes. Thus, we set the following hypotheses to explore the mediation effects of work engagement:

**Hypothesis 4.1.** 
*Work engagement mediates the association between nursing practice environment and strength-oriented care attitudes (Figure 1a).*


**Hypothesis 4.2.** 
*Work engagement mediates the association between nursing practice environment and mental health (Figure 1b).*


**Hypothesis 4.3.** 
*Work engagement mediates the association between nursing practice environment and intention to leave current workplace (Figure 1c).*


**Hypothesis 4.4.** 
*Work engagement mediates the association between nursing practice environment and intention to resign from their profession as a nurse (Figure 1d).*


## 2. Materials and Methods

### 2.1. Design

The current study was a cross-sectional study with quantitative analysis including hypotheses testing.

### 2.2. Participants and Settings

Participants consisted of psychiatric nurses from three psychiatric hospitals in the Kinki and Kyushu region of Japan, which were selected through convenience sampling. The inclusion criteria for psychiatric nurses were as follows: (a) registered or licensed practical nurses, (b) nurses directly engaged in psychiatric patient care, and (c) nurses working in wards or community care departments such as outpatient, home-visiting nursing stations, and psychiatric daycare. The nursing director and nursing deputy director were excluded because they mainly engaged in hospital management. This study had no restrictions on the participants by sociodemographic status (e.g., age, gender, or years of experience).

### 2.3. Procedures

After obtaining permission from the nursing directors, the investigators explained the purpose of this study, research methods, the principle of anonymity, and voluntary participation to all nursing directors and head nurses. Anonymous self-administered questionnaires and research description documents were subsequently distributed to all eligible nurses (*n* = 512) through head nurses at each department. When we distributed the questionnaires to the participants, we asked the head nurses to inform the participants that their cooperation in this study should be of their own free will. It took 20 to 30 min to complete the questionnaire forms, which was clearly stated in the description document. Completed questionnaires were placed into envelopes, sealed, and posted to a collection box located at the departments. Data collection took place from July to August 2021, which was during the COVID-19 pandemic.

### 2.4. Measurement

#### 2.4.1. Work Engagement

Work engagement was assessed using the 9-item short version of the Utrecht Work Engagement Scale (UWES-9) [29,30]. This scale is composed of three subscales measuring: vigor (three items; e.g., “At my work, I feel like I am bursting with energy”), dedication (three items; e.g., “My job inspires me”), and absorption (three items; e.g., “I feel happy when I am working intensely”). It is a 7-point Likert scale ranging from 0 (never) to 6 (always), with higher scores indicating higher work engagement. The average scores of UWES-9 are calculated as an index of work engagement. The Japanese version of the UWES-9 was developed through the process of forward and back translation [30]. It showed good reliability, i.e., good internal consistency (Cronbach α = 0.92) and the test–retest reliability as well as good construct validity [30].

#### 2.4.2. Job Crafting

Job crafting was assessed using Job Crafting Scale developed by Sekiguchi and his colleagues (2017) based on the conceptualization by Wrzesniewski and Dutton (2001) [20,31]. This scale was developed by dual-panel translation methods [31]. It comprises nine items assessing three subscales: task crafting (three items; e.g., “Adding or reducing tasks so that my job can be performed more smoothly”), relational crafting (three items; e.g., “Actively interacting with people through my job”), and cognitive crafting (three items; e.g., “Reframing my job as significant and meaningful”). It is a 7-point Likert scale ranging from 1 (strongly disagree) to 7 (strongly agree). Each subscale score is calculated by dividing the sum of item scores by the number of the items, with higher scores indicating more job crafting behaviors. This scale was found to have acceptable internal consistency (Cronbach α = 0.67–0.80) and good construct validity [31].

#### 2.4.3. Nursing Practice Environment

Nursing practice environment was assessed using the Japanese version of the Practice Environment Scale of the Nursing Work Index with a 4-point Likert scale ranging from 1 (strongly disagree) to 4 (strongly agree) [23,32]. A higher score indicates a more positive nursing practice environment. It consists of the following five subscales: (a) nurse participation in hospital affairs (nine items; e.g., “Opportunity for staff nurses to participate in policy decisions”), (b) manager’s ability, leadership, and support (five items; e.g., “A nurse manager who is a good manager and leader”), (c) staffing and resource adequacy (four items; e.g., “Enough registered nurses to provide quality patient care”), (d) collegial nurse–physician relations (three items; e.g., “Physicians and nurses have good working relationships”), and I nursing foundation for quality of care (ten items; e.g., “Active staff development or continuing education programs for nurses”). The average scores of each subscale are calculated as an index of each component of the nursing practice environment. This scale was developed through the process of forward and back translation procedure [32]. Each subscale was found to have acceptable reliability (Cronbach α = 0.76–0.86 and correlation coefficients based on the test–retest method = 0.65–0.83). Additionally, this scale showed acceptable validity and convergent validity [32].

#### 2.4.4. Strength-Oriented Care Attitudes

Strength-oriented care attitudes were assessed using a scale developed in Japan by Niekawa and his colleagues (2012) [33]. It comprises eleven items, which are composed of three subcomponents measuring the person-centered approach (three items; e.g., “I actively have conversations with patients about their characters and values as well as their illnesses and symptoms”), shared decision making (three items; e.g., “Patients are involved in the conferences regarding goal setting and care planning”), and strength approach (three items; e.g., “I try to discover patient’s personal and environmental strengths through dialogue and involvement”). A higher score means higher strength-oriented care attitudes. All items are measured on a 4-point Likert scale ranging from 0 (rarely do) to 3 (do), and the average scores of this scale are calculated as an index of strength-oriented care attitudes. This scale was confirmed to have generally good convergent validity (Pearson’s correlation coefficient with the Recovery Attitude scale = 0.12–0.23, *p* < 0.05), test–retest reliability (ICC: 0.76–0.84), and internal consistency (Cronbach’s alpha coefficient: 0.65–0.87) [33].

#### 2.4.5. Mental Health of Psychiatric Nurses

Mental health was assessed using the Japanese version of the 5-item World Health Organization Well-Being Index (WHO-5-J). The scale is among the most widely used questionnaires assessing mental health [34,35]. All items are measured on a 6-point Likert scale ranging from 0 (At no time) to 5 (All of the time). The total scores of WHO-5-J are calculated as an index of mental health. Higher scores indicate better mental health, and the total scores below 13 indicate low mental health status. The WHO-5-J was developed through forward and back translation procedures. It showed good reliability (Cronbach α = 0.89) and good construct validity [36,37].

#### 2.4.6. Turnover Intention

Turnover intention was asked by two questions: “intention to leave current workplace” and “intention to resign [from their] profession as a nurse”, with reference to previous research [38,39]. Participants referred to the last 6 months when they responded to each question without considering the influence of the COVID-19 pandemic. Each item was rated on a 5-point Likert scale ranging from 1 (Not at all) to 5 (Very strongly). Higher scores indicated stronger turnover intention.

#### 2.4.7. Demographic Variables

Demographic variables included age, gender, years of nursing experience, years of work tenure in psychiatric or mental health services, qualification, job position, employment status, educational level, and settings.

### 2.5. Statistical Analysis

We conducted a descriptive analysis of the study variables by using the software SPSS version 25 for Windows (IBM Corp., Armonk, NY, USA). Student’s t-test was performed to compare mean work engagement scores between subgroups of discrete variables. Pearson’s correlation coefficients were calculated to determine correlations between continuous variables.

We conducted structural equation modeling (SEM) to test our hypothetical model using the Amos version 26.0. software (IBM Corp., Armonk, NY, USA). In order to avoid overly complex models, SEM was performed for each of the four outcome variables: strength-oriented care attitudes (Model 1), mental health (Model 2), intention to leave current workplace (Model 3), and intention to resign from their profession as a nurse (Model 4). Bootstrap resampling with 1000 bootstrap samples was also performed to estimate confidence intervals. The bias-corrected confidence intervals were reported in this study. Latent constructs were identified by the observed indicators using parcels, as suggested by earlier studies [40,41]. Work engagement, job crafting, nursing practice environment, and strength-oriented care attitudes were parceled into each subcomponent (e.g., work engagement was parceled into its three factors (vigor, dedication, and absorption). Mental health was parceled into two elements through the division of odd and even numbered items of WHO-5-J. Moreover, each turnover intention variable (i.e., intention to leave current workplace and intention to resign from their profession as a nurse) was set with a path coefficient of 1 and an error variance value of 0.5.

Alongside the chi-square (χ^2^) statistic, we used the following fit indices to evaluate the model: the goodness-of-fit index (GFI), the adjusted goodness-of-fit index (AGFI), the Comparative Fit Index (CFI), the normed fit index (NFI), and the root mean square error of approximation index (RMSEA). A satisfying model needs to meet the following criteria: GFI > 0.90, AGFI > 0.90, CFI > 0.90, NFI > 0.90, and RMSEA < 0.05 [42]. Nevertheless, RMSEA < 0.08 was also acceptable, which has been recognized by researchers [43]. *p* values of less than 0.05 were considered statistically significant. All the tests were two-tailed.

### 2.6. Ethical Considerations

Participants received a written explanation of the aims, methods, voluntary nature of the study, and the protection of anonymity. Only those who agreed to participate in the study answered the questionnaire. The aims and procedures of this study were approved by the Institutional Review Board of the Graduate School of Health Science, Kobe University (No. 1009).

## 3. Results

Among 512 psychiatric nurses who received questionnaires, 325 returned the questionnaires (response rate: 63.5 %). However, 16 nurses were excluded due to missing responses regarding one or more items of the variables used in the main analysis. We used data from the remaining 309 participants to conduct the analyses in this study (valid response rate: 60.4%).

### 3.1. Participants’ Characteristics

Table 1 shows the characteristics of the participants. The mean age of the participants was 43.3 years (range: 20–65 years; SD = 10.7), and 54.7% were female. The average years of nursing experiences was 17.8 years (range: 0.3–44.3 years; SD = 10.6). Most of the psychiatric nurses were full-time (*n* = 295; 95.5%), registered nurses (*n* = 304; 98.4%), staff nurses (*n* = 239, 77.4%), and worked in inpatient psychiatric wards (*n* = 251; 81.2%).

### 3.2. Scores of Each Scale and Correlations between Scores for Each Scale

Table 2 shows the means, standard deviations, Cronbach’s α coefficients, and correlations of all scales included in this study. A mean score of work engagement among the total participants was 2.58 (SD = 1.01). The mean score of WHO-5-J in this study was 11.85 (SD = 5.25). No significant differences in work engagement were observed by study variables except for settings (i.e., wards/community care settings). Student’s t-test showed that the work engagement score of nurses in wards (M = 2.52, SD = 1.03) was significantly lower than those in community care settings (M = 2.85, SD = 0.92; t = −2.22, *p* = 0.03).

Cognitive crafting had a stronger positive relationship with work engagement than other variables (r = 0.59, *p* < 0.01) (Table 2).

### 3.3. Hypothesis Testing

The results of the tests of hypothesized models regarding the antecedents and outcomes of work engagement among psychiatric nurses are shown in Figure 2a–d. Acceptable model fits were shown in all four models (Table 3). Additional analyses were completed by controlling for gender, job position, and settings (ward settings or community care settings) as potential confounders. The path coefficients remained virtually the same as those of all models in Figure 2, while the model fit was slightly lower (e.g., Model 1 with control variables had the following model fit indices; χ^2^ = 245.56, df = 108, GFI: 0.91, AGFI: 0.88, CFI: 0.94, NFI: 0.90, RMSEA: 0.07). The impact of the control variables on the model was weak. None of the control variables significantly affected the structural paths in the model (*p*’s > 0.05).

#### 3.3.1. Antecedents of Work Engagement

The results of structural equation modeling for all four models showed that job crafting (e.g., in Model 1; β = 0.57, 95%CI; [0.44–0.69], *p* < 0.01) and nursing practice environment (e.g., in Model 1; β = 0.23, 95%CI; [0.07–0.36], *p* = 0.01) had direct and positive effects on work engagement (Figure 2). Moreover, job crafting had an indirect and positive effect on work engagement mediated by nursing practice environment (e.g., in Model 1; indirect effect = 0.04, 95%CI [0.02, 0.08], *p* = 0.01). Figure 2a–d showed the results regarding the direct and indirect effects of antecedents on work engagement for each model. Thereby, H1-a, H1-b, and H2 were supported.

#### 3.3.2. Outcomes of Work Engagement

Work engagement had direct and positive effects on strength-oriented care attitudes (β = 0.15, 95%CI; [0.01, 0.29], *p* = 0.04) and mental health (β = 0.37, 95%CI; [0.23, 0.47], *p* < 0.01). Work engagement also had a direct and negative association with intention to resign from their profession as a nurse (β = −0.17, 95%CI; [−0.31, −0.04], *p* = 0.01), while it did not have a direct and significant association with intention to leave current workplace (β = −0.08, 95%CI; [−0.2, 0.07], *p* = 0.30). Thus, H3-a, H3-b, and H3-d were supported, but H3-c was not supported.

In addition, the nursing practice environment had direct and significant effects on strength-oriented care attitudes (β = 0.34, 95% CI; [0.21, 0.46], *p* < 0.01), mental health (β = 0.36, 95% CI; [0.24, 0.48], *p* < 0.01), intention to leave current workplace (β = −0.50, 95% CI; [−0.65, −0.36], *p* < 0.01), and intention to resign from their profession as a nurse (β = −0.36, 95% CI; [−0.52, −0.19], *p* < 0.01).

#### 3.3.3. Mediating Effects of Work Engagement on Outcome Variables

Work engagement partly mediated the effects of nursing practice environment on strength-oriented care attitudes (0.03, 95%CI; [0.003, 0.07], *p* = 0.02), mental health (0.07, 95%CI; [0.03, 0.12], *p* < 0.01), and intention to resign from their profession as a nurse (−0.03, 95%CI; [−0.08, −0.01], *p* = 0.01), while it did not mediate the relationship between nursing practice environment and intention to leave current workplace (−0.01, 95%CI; [−0.05, 0.01], *p* = 0.2). Thus, H4-a, H4-b, and H4-d were supported, but H4-c was not supported.

## 4. Discussion

This study highlighted the possible antecedents (i.e., job crafting and nursing practice environment) and outcomes (i.e., strength-oriented care attitudes, mental health, and turnover intention) of work engagement among psychiatric nurses. Notably, job crafting and nursing practice environment had direct/indirect and positive effects on work engagement. Especially, job crafting had a greater impact on work engagement. Work engagement also had positive associations with strength-oriented care attitudes and mental health, while the effect of work engagement on strength-oriented care attitudes was small. In addition, work engagement had a negative relationship with intention to resign their profession as a nurse, while work engagement had no significant relationship with intention to leave their current workplace.

Regarding outcomes of work engagement, our study is the first to demonstrate the significant effects of work engagement on strength-oriented care attitudes, although the correlation was quite weak. Highly engaged nurses usually exhibit more energy and experience positive emotions at work [27]. Positive feelings toward patients can let nurses focus on patients’ strengths and potential rather than on their problems and shortcomings. Consequently, it would lead to a positive outlook toward patients’ recovery as well as better strengths-oriented care attitudes. These in turn might contribute to high-quality mental health care [8,44], including discharge support and care for realizing community life patients want [33]. Therefore, work engagement might be an important psychological driver for practicing strength-oriented care in the field of psychiatric nursing, where nurses sometimes have difficulty in having positive feelings due to negative experiences such as violence from patients. As this study was conducted during the COVID-19 pandemic, some difficulties might arise, due to the restrictions in patient involvement and discharge support in terms of infection prevention [45,46]. Since it may be difficult for psychiatric nurses to implement care plans based on strength orientation under these circumstances, the association between strength-oriented care attitudes and work engagement may have been weakened in this study.

Work engagement also had a positive impact on mental health among psychiatric nurses, which is consistent with the findings of previous studies among general nurses [25,47]. Highly engaged nurses may tend to have good mental health by fulfilling their needs for self-actualization through their work. This is a result of nurses easily having positive work-related emotions, despite sometimes facing difficulties at work. Additionally, COVID-19 causes nurses psychological distress and exacerbates depressive symptoms [46]. In fact, many participants in this study were in poor mental health conditions, as the average WHO-5-J scores were below 13. The psychological distress caused by the COVID-19 pandemic might weaken the association between work engagement and mental health. However, given our findings that showed the positive relationship between two variables even under these circumstances, work engagement may help psychiatric nurses maintain and improve their mental health.

The results in this study also suggested that work engagement could contribute to lower intention to resign their profession as a nurse. However, work engagement did not show a significant relationship with intention to leave their current workplace. Several previous studies have reported a negative association between work engagement and turnover intention [15,48]. However, these studies analyzed turnover intention as a single concept without distinguishing between intention to resign from their profession as a nurse and intention to leave their current workplace. The results of our study suggest that work engagement may have different effects on two types of turnover intention, respectively. Future research distinguishing these two types of turnover intentions and examining their relationship to work engagement may provide useful suggestions for preventive measures against nursing shortages and turnover from workplace.

Among antecedents of work engagement, job crafting had a direct/indirect and positive effect on psychiatric nurses’ work engagement. This was consistent with the findings of previous studies for general nurses [49,50,51]. Among three job-crafting components, cognitive crafting had the strongest correlation with work engagement. Cognitive crafting is related to higher self-efficacy by reframing one’s work as meaningful or redefining the purpose and the meaning of their work [21]. As the self-efficacy of psychiatric nurses tends to be low due to difficulties in mental care and work-related stressors [52], cognitive crafting can play a significant role to enhance psychiatric nurses’ self-efficacy, resulting in facilitating work engagement.

Nursing practice environment also had a direct and positive effect on work engagement. In addition, nursing practice environment was directly and positively associated with strength-oriented care attitudes and mental health as well as directly and negatively related with turnover intention (i.e., intention to resign from their profession as a nurse and intention to leave current workplace). In mental health care, psychiatric nurses carefully observe patients’ conditions including psychiatric symptoms to notice any changes and empower them to draw on their inner resources (e.g., their hopes and strength) through a positive therapeutic relationship [53,54]. A healthier nursing practice environment including collaborative relationships with physicians, plenty of support from supervisors, rich educational opportunities, and abundant human resources will help psychiatric nurses efficiently perform these professional tasks. Additionally, such a healthy nursing practice environment may contribute to not only the enhancement of their ability as psychiatric nurses but also the improvement in the difficulties they face regarding nurse–patient relationships. These work-related positive experiences may encourage them to engage in psychiatric nursing, ensuing the improvement of work-related outcomes. Given that the findings of this Japanese study were consistent with past studies in Western countries and other Asian countries [15,16,55], healthier nursing practice environment may be a key predictor for work engagement globally.

The results of this study should be interpreted with caution due to limitations. First, participants were selected from three psychiatric hospitals through convenience sampling, and psychiatric nurses at clinics or general hospitals were not included. Moreover, the majority of the participants were from wards. Since the work engagement score of nurses in wards was significantly lower than those in community care settings, generalizability should be confined. Second, since we used a cross-sectional design, causal relationships between work engagement and related factors were not confirmed. Future studies with longitudinal design among psychiatric nurses in various settings, including more community care settings, are needed. As this study was conducted during the COVID-19 pandemic with a specific stressful situation, further studies may reveal whether the result in this study is replicable after the pandemic.

### Implications for Practice

This study suggested that work engagement is important for psychiatric nurses and nursing managers to improve work-related outcomes (i.e., strength-oriented care attitudes, mental health, and turnover intention). To enhance work engagement among psychiatric nurses, it may be important to incorporate job crafting (especially, cognitive crafting) into clinical practice and to improve nursing practice environment such as increasing supervisor’s support, making harmonious relationships with physicians and improving the nursing foundations for quality of care. One strategy to introduce cognitive crafting into practice may be to set educational opportunities for psychiatric nurses to re-think and reflect on the benefit and value of their work.

## 5. Conclusions

Job crafting and healthier nursing practice environment could enhance work engagement among psychiatric nurses. Higher work engagement as well as healthier nursing practice environment could help to improve work-related outcomes (i.e., strength-oriented care attitudes, mental health, and turnover intention), which may secondarily improve quality of mental health care.

## Figures and Tables

**Figure 1 healthcare-11-00295-f001:**
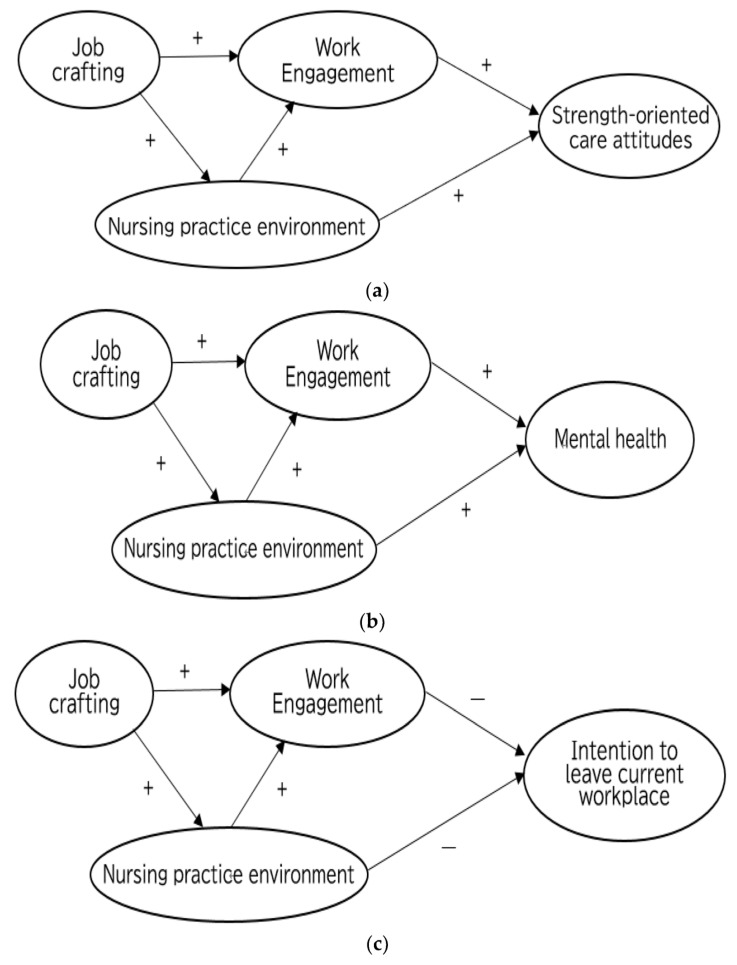
(**a**): Hypothesized model (Model 1) for antecedents and outcome (strength-oriented care attitudes) of work engagement among psychiatric nurses. (**b**): Hypothesized model (Model 2) for antecedents and outcome (mental health) of work engagement among psychiatric nurses. (**c**): Hypothesized model (Model 3) for antecedents and outcome (intention to leave current workplace) of work engagement among psychiatric nurses. (**d**): Hypothesized model (Model 4) for antecedents and outcome (intention to resign from their profession as a nurse) of work engagement among psychiatric nurses. Note: + means the hypothesized relationship is positive, − means the hypothesized relationship is negative.

**Figure 2 healthcare-11-00295-f002:**
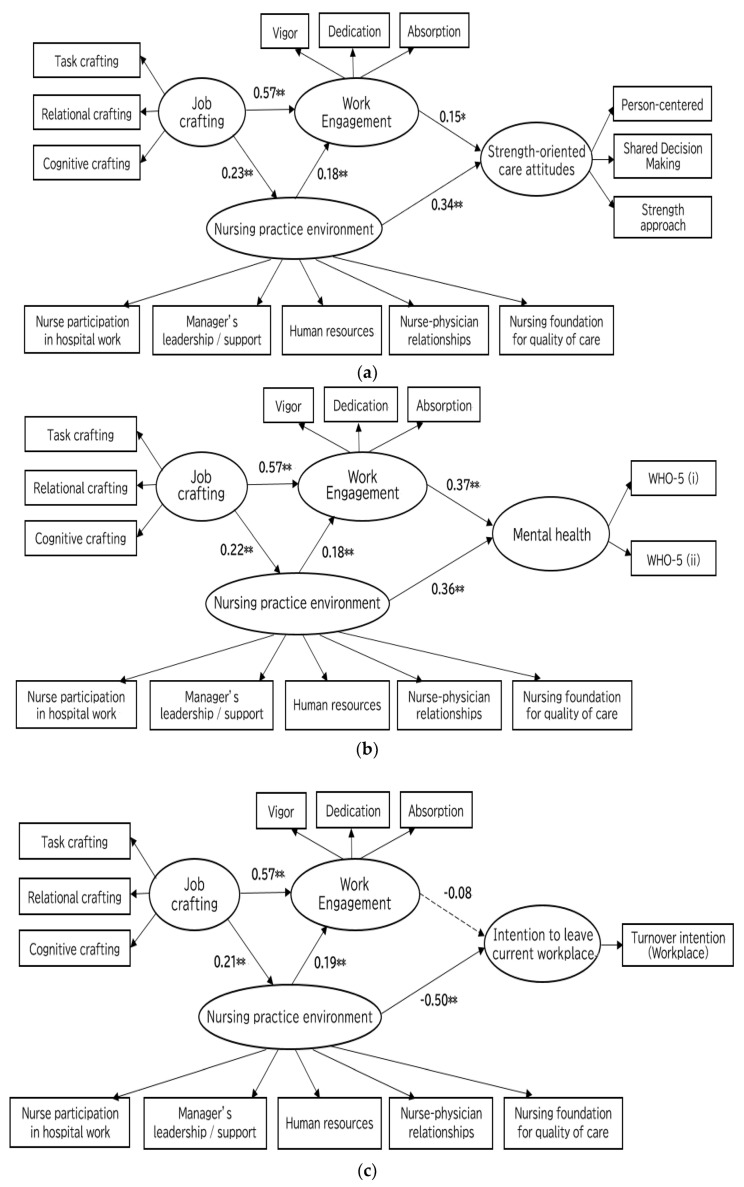
(**a**). The result of Model 1 regarding antecedents and outcome (strength-oriented care attitudes) of work engagement among psychiatric nurses. Note: * *p* < 0.05, ** *p* < 0.01. Indirect effect of job crafting on work engagement mediated by nursing practice environment; 0.04, 95%CI [0.02, 0.08], *p* = 0.01. (**b**). The result of Model 2 regarding antecedents and outcome (mental health) of work engagement among psychiatric nurses. Note: * *p* < 0.05, ** *p* < 0.01. WHO-5 (i) means the total score of the odd-numbered items (i.e., items 1, 3, and 5) on the WHO-5-J. WHO-5 (ii) means the total score of the even-numbered items (i.e., items 2 and 4) on the WHO-5-J. Indirect effect of job crafting on work engagement mediated by nursing practice environment; 0.04, 95%CI [0.01, 0.08], *p* = 0.01. (**c**). The result of Model 3 regarding antecedents and outcome (intention to leave current workplace) of work engagement among psychiatric nurses. Note: * *p* < 0.05, ** *p* < 0.01. Dotted line represents nonsignificant path (*p* > 0.05). Indirect effect of job crafting on work engagement mediated by nursing practice environment; 0.04, 95%CI [0.01, 0.08], *p* = 0.01. (**d**). The result of Model 4 regarding antecedents and outcome (intention to resign from their profession as a nurse) of work engagement among psychiatric nurses. Note: * *p* < 0.05, ** *p* < 0.01. Indirect effect of job crafting on work engagement mediated by nursing practice environment; 0.04, 95%CI [0.01, 0.08], *p* = 0.01.

**Table 1 healthcare-11-00295-t001:** Demographics of the participants (*n* = 309).

Sociodemographic Variables	Mean ± SD [Range] or Number (%)
Age	43.3 ± 10.7 [20–65]
Gender	
Male	133 (43.0)
Female	169 (54.7)
Unknown	7 (2.3)
Years of nursing experiences	17.8 ± 10.6 [0.3–44.3]
Years of work tenure in psychiatric or mental health services	13.7 ± 9.9 [0.2–44.3]
Qualification	
Registered nurses	304 (98.4)
Licensed practical nurses	3 (1.0)
Unknown	2 (0.6)
Job Position	
Head nurse	22 (7.1)
Deputy head	13 (4.2)
Chief nurse	35 (11.3)
Staff nurse	239 (77.4)
Employment status	
Full-time	295 (95.5)
Part-time	12 (3.9)
Unknown	2 (0.6)
Education level	
Four-year college degree or above	68 (22.0)
Lower than college degree	233 (75.5)
Unknown	8 (2.5)
Settings	
Ward settings	251 (81.2)
Acute psychiatric ward	106 (34.4)
Chronic psychiatric ward	107 (34.5)
Medical treatment and supervision ward	21 (6.8)
Child adolescent psychiatric ward	12 (3.9)
COVID-19-related ward	5 (1.6)
Community care settings	57 (18.5)
Outpatient	15 (4.9)
Home-visit nursing station	29 (9.4)
Psychiatric daycare	13 (4.2)
Unknown	1 (0.3)

Note: SD; Standard Deviation.

**Table 2 healthcare-11-00295-t002:** Pearson’s correlations coefficients, means, and standard deviations of the variables in this study (*n* = 309).

	1	2	3	4	5	6	7	8	9	10	11	12	13
1 Task crafting ^§1^	1												
2 Relational crafting ^§1^	0.58 ^**^	1											
3 Cognitive crafting ^§1^	0.47 ^**^	0.60 ^**^	1										
4 Participation opportunity in work ^§2^	0.15^*^	0.15 ^**^	0.25 ^**^	1									
5 Supervisors’ leadership and support ^§2^	0.04	0.10	0.18 ^**^	0.52 ^**^	1								
6 Human resources ^§2^	0.06	0.08	0.07	0.46 ^**^	0.38 ^**^	1							
7 Nurse–physician collaborations ^§2^	0.08	0.13 ^*^	0.24 ^**^	0.39 ^**^	0.35 ^**^	0.40 ^**^	1						
8 Foundation for quality of care ^§2^	0.02	0.07	0.13 ^*^	0.63 ^**^	0.50 ^**^	0.61 ^**^	0.46 ^**^	1					
9 Strength-oriented care attitudes ^§3^	0.32 ^**^	0.30 ^**^	0.29 ^**^	0.29 ^**^	0.16 ^**^	0.28 ^**^	0.13 ^*^	0.32 ^**^	1				
10 Mental health ^§4^	0.20 ^**^	0.34 ^**^	0.32 ^**^	0.30 ^**^	0.31 ^**^	0.34 ^**^	0.32 ^**^	0.36 ^**^	0.28 ^**^	1			
11 Intention to leave current workplace	−0.04	−0.13 ^*^	−0.11 ^*^	−0.31 ^**^	−0.28 ^**^	−0.37 ^**^	−0.25 ^**^	−0.34 ^**^	−0.09	−0.32 ^**^	1		
12 Intention to resign from their profession as a nurse	−0.10	−0.13 ^*^	−0.12 ^*^	−0.25 ^**^	−0.21 ^**^	−0.28 ^**^	−0.23 ^**^	−0.24 ^**^	−0.08	−0.39 ^**^	0.76 ^**^	1	
13 Work engagement ^§5^	0.30 ^**^	0.47 ^**^	0.59 ^**^	0.32 ^**^	0.18 ^**^	0.14 ^*^	0.16 ^**^	0.18 ^**^	0.23 ^**^	0.42 ^**^	−0.17 ^**^	−0.21 ^**^	1
[Min, Max]	[1, 7]	[1, 7]	[1, 7]	[1, 4]	[1, 4]	[1, 4]	[1, 4]	[1.1, 4]	[0, 3]	[0, 25]	[1, 5]	[1, 5]	[0, 5.56]
Mean	4.67	4.60	4.53	2.45	2.91	2.40	2.80	2.62	1.69	11.85	2.25	1.27	2.58
SD	1.05	1.03	1.14	0.51	0.62	0.66	0.51	0.51	0.50	5.25	2.02	1.20	1.01
Cronbach’s α coefficient in this study	0.81	0.83	0.87	0.84	0.92	0.87	0.85	0.88	0.88	0.93	-	-	0.93

Note: ^§1^: Job Crafting Scale, ^§2^: Practice Environment Scale of the Nursing Work Index, ^§3^: The scale for strength-oriented care attitudes, ^§4^: The Japanese version of the 5-item World Health Organization Well-Being Index, ^§5^: The 9-item short version of the Utrecht Work Engagement Scale. * *p* < 0.05, ** *p* < 0.01. SD; Standard Deviation. Cronbach’s α coefficients for intention to leave current workplace and intention to resign from their profession as a nurse were not calculated due to single items.

**Table 3 healthcare-11-00295-t003:** Fit indices for all hypothetical models.

Models	χ^2^	df	GFI	AGFI	CFI	NFI	RMSEA
Model 1	176.38	72	0.93	0.89	0.95	0.93	0.07
Model 2	156.59	60	0.93	0.89	0.96	0.93	0.07
Model 3	120.73	50	0.94	0.90	0.96	0.94	0.07
Model 4	125.09	50	0.94	0.90	0.96	0.93	0.07

Note: Models 1–4 tested the relationships among antecedents, work engagement, and outcomes. Strength-oriented care attitudes (Model 1); mental health (Model 2); intention to leave current workplace (Model 3); and intention to resign from their profession as a nurse (Model 4).

## Data Availability

The data that support the findings of this study are available from the corresponding author, R.C., upon reasonable request.

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
