# Peer review of "Antecedents and Outcomes of Work Engagement among Psychiatric Nurses in Japan"

_healthcare, 2023, doi:10.3390/healthcare11030295_

Round 1

Reviewer 1 Report

Authors prepared the manuscript and written systematically. However, to improve the quality of manuscript, some recommendation below need to be addressed:

1. Figure 1a-d are missed some word in the right side

2. Methods: Please describe instrument development appropriately (translation, validity process, reliability measurement) for work enggagement, job crafting, nursing practice environment, strength-oriented care attitude, mental health, turn over intention. Also, please present these instrument in systematic table to make easier for the readers

3. Please stated how the questionnaire was delivered to the respondent and how it take long

4. Line 325: Regarding outcomes of work engagement, our study is the first to demonstrate the significant effects of work engagement on strength-oriented care attitudes, although the correlation was quite weak...please explain why the correlation between two variables are weak? is is reflect the real situation? please discuss

5. Line 343: work engagement did not show a significant relationship with intention to leave their current workplace....please elaborate with other previous studies. 

6. Line 383: As this study was conducted during the COVID-19 pandemic with a specific stressful situation, further studies may reveal whether the result in this study is replicable after the pandemic.....please add more discussion about this situation and how it have impact on study results, in the discussion on mental health or other variable. 

Author Response

Reply to Reviewer 1

First, we appreciate your comments and useful advice on our manuscript. We carefully revised our manuscript considering the following points.

Reviewer’s Comment #1

Figure 1a-d are missed some word in the right side.

Thank you for your helpful advice. We carefully modified the Figure 1a-d to read.

Reviewer’s Comment #2

Methods: Please describe instrument development appropriately (translation, validity process, reliability measurement) for work engagement, job crafting, nursing practice environment, strength-oriented care attitude, mental health, turn over intention. Also, please present these instruments in systematic table to make easier for the readers.

Thank you for your valuable advice. We added the explanation of the instrument development in Methods section (i.e., translation process, as well as validity and reliability information) (L.165-168, 172, 194-197, 209-212), except for turnover intention, which was asked by two questions with reference to previous research.

We also considered to make systematic table of the instruments used in this study, based on your advice. However, as we found it difficult to describe the information in a table, due to limited width of the paper, we eventually decided to elaborate the information about scales in the text. We hope it is still acceptable. Once again, we appreciate your important advice.

Reviewer’s Comment #3

  1. Please state how the questionnaire was delivered to the respondent and how it takes long.

We totally agree with your advice. We added the following explanations in 2.3 Procedures section (L.150-154):

“When we distributed the questionnaires to the participants, we asked the head nurses to inform the participants that their cooperation in this study should be by their own free will. It took 20 to 30 minutes to complete the questionnaire forms, which was clearly stated in the description document.”

Reviewer’s Comment #4

  1. Line 325: Regarding outcomes of work engagement, our study is the first to demonstrate the significant effects of work engagement on strength-oriented care attitudes, although the correlation was quite weak...please explain why the correlation between two variables are weak? Does it reflect the real situation? please discuss.

Thank you for your significant advice. We added a possible explanation of the weak association between two variables as below:

“As this study was conducted during the COVID-19 pandemic, some difficulties might happen, due to the restrictions in patient involvement and discharge support in terms of infection prevention [45, 46]. Since it may be difficult for psychiatric nurses to implement care plans based on strength orientation under these circumstances, the association between strength-oriented care attitudes and work engagement may have been weakened in this study.” (L.354-359)

Reviewer’s Comment #5

  1. Line 343: work engagement did not show a significant relationship with intention to leave their current workplace....please elaborate with other previous studies.

According to your suggestion, we carefully modified the discussion contents of turnover intention as below:

“Several previous studies have reported a negative association between work engagement and turnover intention [15, 48]. But these studies analyzed turnover intention as a single concept without distinguishing intention to resign their profession as a nurse from intention to leave their current workplace. The results of our study suggest that work engagement may have different effects on two types of turnover intention, respectively. Future research distinguishing these two types of turnover intentions and examining their relationship to work engagement may provide useful suggestions for preventive measures against nursing shortages and turnover from workplace.” (L.374-382)

Reviewer’s Comment #6

  1. Line 383: As this study was conducted during the COVID-19 pandemic with a specific stressful situation, further studies may reveal whether the result in this study is replicable after the pandemic.....please add more discussion about this situation and how it have impact on study results, in the discussion on mental health or other variable.

Thank you for your helpful advice. Based on your advice, we added the discussion on strength-oriented care attitudes and mental health as below:

 “This study was conducted during the COVID-19 pandemic, some difficulties might happen due to restrictions in patient involvement and discharge support in terms of infection prevention [45, 46]. Since it may be difficult for psychiatric nurses to implement care plans based on strength orientation under these circumstances, the association between strength-oriented care attitudes and work engagement may have been weakened in this study.” (L.354-359)

“Additionally, COVID-19 causes nurses psychological distress and exacerbates depressive symptoms [46]. In fact, many participants in this study were in poor mental health conditions, as the average WHO-5-J scores were below 13. The psychological distress caused by the COVID-19 pandemic might weaken the association between work engagement and mental health. However, given our findings that showed the positive relationship between two variables even under these circumstances, work engagement may help psychiatric nurses to maintain and improve their mental health.” (L.364-371)

Moreover, we modified the contents of measurement and results regarding mental health in order to keep consistency throughout the paper.

First, regarding the measurement (WHO-5-J), we added the following sentence:

“Higher scores indicate better mental health and the total scores below 13 indicate low mental health status.” (L. 218-221)

Second, regarding the results of WHO-5-J scores, we added the following explanation:

“The WHO-5-J mean score in this study was 11.85 (SD = 5.25).” (L.282-283)

Finally, we added the following reference for the modification of discussion contents;

  • [45] Bhandari M, Yadav U, Dahal T, & Karki, A. (2022). Depression, Anxiety and Stress among Nurses Providing Care to the COVID-19 Patients: A Descriptive Cross-sectional Study, J Nepal Med Assoc, 60(246), 151-154. https://doi.org/10.31729/jnma.7235
  • [46] Inamata, Y., Furuya, I., Oozeki, H., & Miyabayashi, I. (2022). Impact and current status of visitation restrictions caused by COVID-19 -Using Riessman's thematic analysis-. Seisen Jogakuin College Journal of Nursing, 2(1), 41-56.

http://id.nii.ac.jp/1048/00000574/

  • [48] Kiky, S., & Daniel, L. (2021). Work Engagement and Turnover Intention: The Moderating Effect of Organizational Justice. Advances in Social Science, Education and Humanities Research, 570, 58-65. http://doi.org/10.2991/assehr.k.210805.009.

We hope that the revised manuscript is now acceptable for publication in Healthcare.

Thank you for your helpful advice and consideration.

Reviewer 2 Report

Dear authors,

It is an honor for me to review your paper titled “Antecedents and Outcomes of Work Engagement among Psychiatric Nurses in Japan”.

The study presents a highly relevant research topic, not only in professional nursing practice, but also in the training of future nursing professionals.

Regarding the introduction and theoretical framework, it would be interesting to include the training of nurses, since there is a strong vocational component, so I suggest that you include the following studies:

Sánchez-Bolívar, L., Escalante-González, S., & Martínez-Martínez, A. (2022). Motivation and Social Skills in Nursing Students Compared to Physical Education Students. SPORT TK, 11(5). https://doi.org/10.6018/sportk.462121

Sánchez-Bolívar, L.; Vázquez, L. M. (2022). Emotional Intelligence Of Nursing Degree Students According To Gender And Religion. Journal of Sport and Health Research. 14(Supl 1):35-42.

Available online: https://recyt.fecyt.es/index.php/JSHR/article/view/95389

Regarding the method, the definition of the research design is incomplete. In this sense, the type of study developed, descriptive, relational and interpretative in this case, must be included.

The sample is not clearly defined. Although the inclusion criteria are reflected, the total sample and its description must be reflected, according to gender, age, and other sociometric variables such as years of profession or any other that has been included. Likewise, the sampling error assumed in the study must be reflected.

The rest of the sections are well prepared and are consistent with the entire paper.

Author Response

Reply to Reviewer 2

First, we appreciate your comments and useful advice on our manuscript. We carefully revised our manuscript considering the following points.

Reviewer’s Comment #1

The study presents a highly relevant research topic, not only in professional nursing practice, but also in the training of future nursing professionals.

Regarding the introduction and theoretical framework, it would be interesting to include the training of nurses, since there is a strong vocational component, so I suggest that you include the following studies:

  • Sánchez-Bolívar, L., Escalante-González, S., & Martínez-Martínez, A. (2022). Motivation and Social Skills in Nursing Students Compared to Physical Education Students. SPORT TK, 11(5). https://doi.org/10.6018/sportk.462121
  • Sánchez-Bolívar, L.; Vázquez, L. M. (2022). Emotional Intelligence of Nursing Degree Students According To Gender And Religion. Journal of Sport and Health Research. 14(Supl 1):35-42.

https://recyt.fecyt.es/index.php/JSHR/article/view/95389

Thank you for your helpful advice. We learned a lot from your useful advice.

We read the two previous articles you suggested and these studies were very informative for us. In actual, many empirical studies suggested that work engagement is a different concept from motivation. In addition, target population of this study was not nursing students but psychiatric nurses. From these points, we eventually decided not to cite these articles in the manuscript, considering main focus on this study.

Once again, we appreciate your important advice. We hope you understand it.

Reviewer’s Comment #2

Regarding the method, the definition of the research design is incomplete. In this sense, the type of study developed, descriptive, relational and interpretative in this case, must be included.

The sample is not clearly defined. Although the inclusion criteria are reflected, the total sample and its description must be reflected, according to gender, age, and other sociometric variables such as years of profession or any other that has been included. Likewise, the sampling error assumed in the study must be reflected.

Thank you for your significant advice. As you pointed out, we added the below explanations of research design.

“The current study was a cross-sectional study with quantitative analysis including hypotheses testing.” (L.134-135)

Besides, we confirmed that we adequately described the characteristics of the participants in Table 1.

We also added the sentence “This study had no restrictions for the participants by sociodemographic status (e.g., age, gender, or years of experience).” (L.143-144)

We also explained the limitation of the sampling in the last part of the Discussion section. Given that the valid response rate was higher than 60%, we assume that significant sampling error did not happen in this study.

We hope that the revised manuscript is now acceptable for publication in Healthcare.

Thank you for your consideration.

Round 2

Reviewer 2 Report

Thank you for your work